# SegmentWithSAM: 3D Slicer Extension for Segment Anything Model (SAM)

**Zafer Yildiz**[1]                                        ZAFER.YILDIZ@DUKE.EDU

[1] *Department of Radiology, Duke University, Durham, NC, 27708, USA*

**Hanxue Gu**[2]                                           HANXUE.GU@DUKE.EDU

[2] *Department of Electrical and Computer Engineering, Duke University, Durham, NC, 27708, USA*

**Jikai Zhang**[2]                                         JIKAI.ZHANG@DUKE.EDU

**Jichen Yang**[2]                                         JICHEN.YANG@DUKE.EDU

**Maciej A. Mazurowski**[1,2,3,4]                          MACIEJ.MAZUROWSKI@DUKE.EDU

[3] *Department of Computer Science, Duke University, Durham, NC, 27708, USA*

[4] *Department of Biostatistics & Bioinformatics, Duke University, Durham, NC, 27708, USA*

**Editors:** Accepted for publication at MIDL 2024

## Abstract

The development of reliable automated deep learning-based algorithms for the segmentation of medical images is heavily reliant on training data in the form of images along with outlines of the objects of interest. However, manual annotation of medical images is very time-consuming and often requires highly specialized expertise. Here, we provide software that incorporates the recently developed and highly impactful Segment Anything Model (SAM) into the popular software for the visualization and annotation of medical images, 3D Slicer. SAM has been developed to segment any object with prompt-based user guidance. It has been shown to be successful in aiding some annotations in medical imaging. The software described in this paper allows to leverage the power of SAM while using the highly convenient and publicly available 3D Slicer software. Our code is publicly available on GitHub, and it can be installed directly from the Extension Manager of 3D Slicer.

**Keywords:** Segmentation, SAM, 3D Slicer

## 1. Introduction

The segmentation process is central to maximizing the utility of medical imaging (Patil and Deore, 2013). This intricate procedure involves partitioning medical images to identify and delineate specific anatomical and abnormal regions. In recent years, the field of medical image segmentation has been revolutionized by deep learning algorithms (Hesamian et al., 2019). These algorithms, trained on carefully annotated images, have shown exceptional capability in automating the segmentation process (Zhou et al., 2018)(Swiecicki et al., 2021). However, training of such algorithms is challenged by their dependency on manual, labor-intensive, annotations requiring radiological expertise.

Among the forefront of recent advancements in segmentation algorithms stands the Segment Anything Model (SAM), which is trained on 11 million images with over 1 billion masks and intents to segment any object in image data using deep learning by Meta (Kirillov et al., 2023). Utilizing the potential of foundation models, SAM provides an automated solution that segments any object or structure with remarkable precision when prompts in the form of points or boxes are provided. Previous studies (Mazurowski et al., 2023)have

shown that SAM is capable of good performance on multiple segmentation tasks on medical images.

To improve the efficiency and accessibility of creating segmentation annotations on medical images, we propose integrating SAM into 3D Slicer, an open-source platform renowned for its extensive capabilities in medical image processing (Fedorov et al., 2012). The synergy between SAM and 3D Slicer allows users to quickly create highly accurate segmentation masks with a few prompt points or bounding boxes without additional medical training. By addressing the longstanding challenges associated with manual segmentation processes, our 3D Slicer extension, SegmentWithSAM, could catalyze future foundation models on medical imaging segmentation. Compared with other similar extensions, for example, Segment Any Medical Model (SAMM) (Liu et al., 2023) and TomoSAM (Semeraro et al., 2023), our SegmentWithSAM is a more comprehensive integration of SAM by allowing the users to choose from 3 different masks generated by SAM and producing the embedding files without the need for any program outside of 3D slicer.

## 2. Methodology

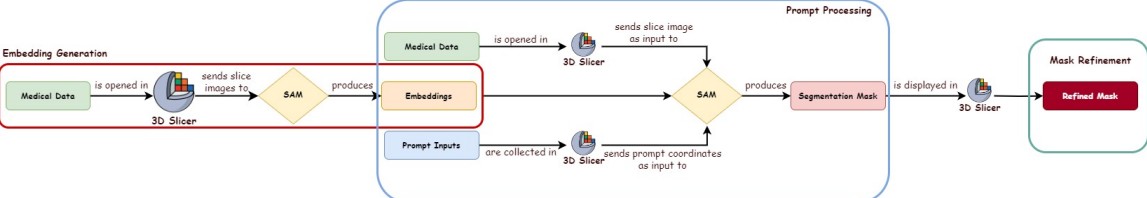

Figure 1: SegmentWithSAM Architecture

SegmentWithSAM aims to communicate between the SAM and the 3D Slicer's user interface (UI). To fulfill this responsibility, it needs the slice images of the medical data opened by the users in 3D Slicer and input prompts given by the users on these slice images. The input prompts can be either positive and negative points, boxes, or even a combination of point and box prompts. SegmentWithSAM converts the prompt inputs to proper format before sending them to SAM, so SAM can process them coherently. After SAM produces the segmentation mask based on the input prompts, SegmentWithSAM reflects the produced mask into the 3D Slicer user interface in almost real-time. This way, the users can see and evaluate the segmentation result in 3D Slicer. If the users are not satisfied with the produced mask, they can change the location or size of the prompt inputs and see the updated segmentation mask produced based on the new prompt input set. By this means, they can determine the most suitable location for the prompt input to get the most accurate mask.

3D slicer is widely used to visualize and process 3D medical images. However, SAM works with 2D image inputs. In order to eliminate this discrepancy, SegmentWithSAM splits 3D medical image volumes into 2D slices and follows a slice-based segmentation strategy. A problem with this slice-based approach is that SAM needs to calculate image embeddings for each slice to segment the input image, which causes the users to wait before segmenting

each slice. To overcome this problem, image embeddings for all slices are produced at once and fully integrated with 3D Slicer before the segmentation process begins.

As a final step, if the users still do not find the segmentation mask that is produced by SAM sufficient enough, they can use the built-in Segment Editor module of 3D Slicer to obtain a further refined version of the segmentation mask by using manual segmentation tools which are provided by 3D Slicer.

## 3. Results

SegmentWithSAM is tested on publicly available medical data sample (Falta et al., 2024) with various prompt input combinations. It is tested using only 1 point prompt (Figure 2 (a)), 2 point prompts (Figure 2 (b)), 1 box prompt (Figure 2 (d)), 2 box prompts (Figure 2 (e)), and a combination of box and point prompts (Figure 2 (f)).

As a result, SegmentWithSAM utilizes all prompt inputs to generate one segmentation mask that tries to bring different prompt inputs together under the same label. SegmentWithSAM also shows better performance when point and box prompts are used together in some scenarios, as can be seen in ((Figure 2 (c) and Figure 2 (f)). To sum up, SegmentWithSAM is able to generate promising segmentation results in various test scenarios.

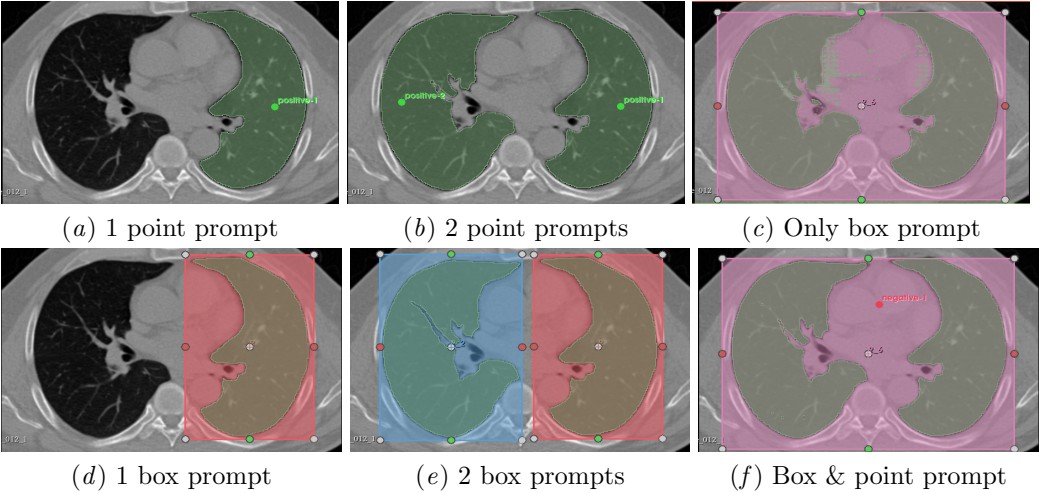

$(a)$ 1 point prompt  $(b)$ 2 point prompts  $(c)$ Only box prompt

$(d)$ 1 box prompt  $(e)$ 2 box prompts  $(f)$ Box & point prompt

Figure 2: Segmentation mask samples based on different prompt input scenarios

## 4. Conclusion

In conclusion, SegmentWithSAM may accelerate the segmentation processes of some medical imaging data. Users can become easily familiar with the usage of the extension since it does not require any coding or deep learning background.

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
