# OpenReview forum: "SegmentWithSAM: 3D Slicer Extension for Segment Anything Model (SAM)"
_MIDL.io/2024/Short_Papers — MIDL 2024 Short Papers_

### Official Review · Reviewer_bKMx · 2024-04-22

**Confidence:** 5
**Final Rating:** 3.5

**Review:**

Summary:

Pros:
1.	The paper provides a practical software that incorporates the recently developed and highly impactful Segment Anything Model (SAM) into the popular software for the visualization and annotation of medical images, 3D Slicer.
2.	The authors have explored the combination effectiveness of different prompt forms.
3.	The authors have released the developed software which will prosper the development of medical image analysis community.

Cons:

1.	The authors should compare the efficiency of annotating with or without the developed software quantitatively, which will greatly advance the paper convincingness.
2.	The paper lacks novelty, since there are also some other works that have done the same work.
3.	The authors have not articulated the integration of SAM and previous 3D slicer. Likewise, the pipeline figure does not showcase that point, where is the previous 3D slicer component.

---

### Decision · Program_Chairs · 2024-04-26

Accept